# A Facile Approach of Fabricating Bifunctional Catalysts for Redox Applications by Uniformly Immobilized Metallic Nanoparticles on NiCr LDH

**DOI:** 10.3390/nano13060987

**Published:** 2023-03-09

**Authors:** Mosaed S. Alhumaimess, Obaid F. Aldosari, Almaha N. Alqhobisi, Laila M. Alhaidari, Afnan Altwala, Linah A. Alzarea, Hassan M. A. Hassan

**Affiliations:** 1Department of Chemistry, College of Science, Jouf University, Sakaka 2014, Saudi Arabia; 2Department of Chemistry, Faculty of Science, Majmaah University, Majmaah 11952, Saudi Arabia

**Keywords:** LDH, CO oxidation, sol immobilization, nitrobenzene reduction, Pd NPs

## Abstract

This study discloses the development of NiCr LDH, Ag@NiCr LDH, and Pd@NiCr LDH bifunction catalysts using a hydrothermal coprecipitation method followed by sol immobilization of metallic nanoparticles. The structures and morphologies of the synthesized nanocomposites were analyzed using FTIR, XRD, XPS, BET, FESEM-EDX, and HRTEM. The catalytic effectiveness of the samples was evaluated by tracking the progression of NaBH_4_-mediated nitrobenzene (NB) reduction to aniline and CO oxidation using UV-visible spectrophotometry and an infrared gas analyzer, respectively. Pd@NiCr LDH displayed much higher performance for both reactions than the bare NiCr LDH. The catalyst Pd@NiCr LDH showed robust catalytic activity in both the oxidation of carbon monoxide (T_50%_ (136.1 °C) and T_100%_ (200.2 °C)) and NaBH_4_-mediated nitrobenzene reduction (98.7% conversion and 0.365 min^−1^ rate constant). The results disclose that the Ni^2+^@ Cr^3+^/Cr^6+^ @Pd° ion pairs inside the LDH act as a charge transfer center and hence significantly enhance the catalytic performance. As a result, this research offers the novel NiCr LDH catalyst as a bifunctional catalyst for air depollution control and the organic transformation process.

## 1. Introduction

Layered double hydroxides (LDHs) have created much attention [1,2,3,4] due to their distinctive two-dimensional (2D) architecture and abundance of active sites for catalysis [5,6,7]. Environmental and clean energy issues can be solved by using the unique properties of fabricated inorganic layered materials with well-defined architecture and distinctive functionalities, such as anionic and cationic clays. Host-guest layered substances are also referred to as anionic clays or LDHs. However, LDH minerals are much less common in nature than cationic clays. LDHs are a significant category of layered composites that are receiving remarkable interest in crucial categories such as catalysis, photocatalysis, adsorbents, photochemistry, biomedical science, electrochemistry, polymerization, magnetization, and environmental implementations [8,9,10,11,12,13]. The majority of LDHs are synthetic materials, and their frameworks resemble those of the naturally occurring mineral hydrotalcite. The low cost, eco-friendliness, layered structure, and high redox activity of LDHs that contain transition metals make them advantageous catalysts [14]. The combination of mixed-valence transition metal ions within the LDH structure enables the formation of various compositions, which can be utilized to create diverse designs for high-performance catalysts [15]. For instance, promising alternatives such as Ni-based LDHs such as Ni/Fe-LDH, Ni/Co-LDH, and Ni/Ti-LDH have been employed [16,17]. According to Ganley et al. [18], Ni-Al LDH has a boosting influence for Pt-based anode cells during the oxidation process that is quite comparable to that using solely bimetallic Pt-Ru anodes. Furthermore, Vlamidis and coworkers [19] showed that Ni/Fe LDHs can reach excellent efficiency in the oxidation process due to the function that Fe plays in the electrocatalytic process. Due to the unique electronic structure of the Cr^3+^ ions, Ni/Cr LDH is anticipated to be a potential bifunctional LDH for oxidation with regard to both performance and durability [20,21,22].

Due to the change in their electronic structure, metallic nanoparticles could display improved physico-chemical characteristics in comparison to their unaltered equivalents. In particular, they have garnered a lot of interest in the area of catalysis [23]. Although substituting non-noble metals can greatly reduce costs, in recent years, their poor catalytic activity has limited their commercial utility. Ag-based catalysts are thought to have the most potential to strike a balance between cost and effectiveness because they are the most affordable noble metal and display superior performance [24]. Furthermore, noble-group metals, particularly Pd, are still the greatest option for redox reactions. The use of precious metals such as palladium, which exhibit robust catalytic properties, has gained significant attention. However, owing to their tiny size and great surface energy, palladium NPs tend to agglomerate, which hinders their catalytic reduction efficiency. One effective approach to prevent particle aggregation is to load the particles onto a carrier substrate. Previously, nitro-compound reduction was accomplished using homogeneous catalysts, which presented several disadvantages, such as challenging separation, inadequate recyclability, and the possibility of contamination from persistent metals in the reaction mixture. To address these problems, considerable attempt was directed towards developing catalysts in which Pd NPs are loaded onto a solid carrier [25].

Researchers recently have concentrated on the structure–activity correlation of Pd-based nanocatalysts, comprising the distribution oxidation state, size distribution, and support impact. As a result, significant effort has been made to create quick and effective preparative approaches for creating metallic nanomaterials with a distinct shape and size. Hence, one of the most interesting ideas for advancing the applicability of catalysis is the design of more affordable, reliable, and high-activity catalysts for redox processes. We believed that the immobilization of Pd or Ag metallic nanoparticles on NiCr LDH would create extremely engaged hybrid materials for catalytic implementations. Ni-Cr LDHs are a type of layered substance comprising of positively charged metal hydroxide layers separated by interlayer anions. The NiCr LDH structure can be described as follows: in the hydroxide layers, the cationic metals fill octahedral positions, while the interlayer anions are coordinated to the metal cations. The location of the metal (Ag or Pd) over NiCr LDH refers to the position of the metal on the surface of the material. In a bifunctional catalyst system, the metal (Ag or Pd) is often used to modify the properties of the NiCr LDH, such as its electronic structure, surface area, and reactivity. One approach to constructing bifunctional catalysts using NiCr LDHs is to load the metal M on the surface of the material, creating a (Ag or Pd) @NiCr LDH composite. This can be achieved through various methods, such as physical adsorption or chemical deposition. In the (Ag or Pd) @NiCr LDH composites, the metal M is located on the surface of the material and is in close proximity to the active sites of the NiCr LDH. This allows the metal (Ag or Pd) to interact with the active sites and modulate their properties, resulting in a bifunctional catalyst with improved performance. Therefore, the location of the metal (Ag or Pd) over NiCr LDH plays a significant role in designing the whole properties and performance of the bifunctional catalyst system [26].

Herein, Ag and Pd NPs immobilized NiCr LDH nanocomposites were successfully synthesized using a reasonably simple approach. The resultant nanocatalysts were examined employing Fourier-transform infrared spectroscopy (FT-IR), X-ray diffraction (XRD), X-ray photoemission spectroscopy (XPS), nitrogen isotherms and pore size distribution, transmission electron microscopy (TEM), and scanning electron microscopy (SEM). After being characterized, the Ag@NiCr LDH and Pd@NiCr LDH nanocomposites were exploited as catalysts for the nitrobenzene reduction and CO oxidation processes. In fact, both of these reactions (redox reactions) are crucial for the organic transformation process which is extremely valuable in industrial chemistry, whereas the CO oxidation reaction is extremely important in air pollution control [27,28].

## 2. Materials and Methods

### 2.1. Materials

The chemicals chromium (III) nitrate (99%, Sigma-Aldrich, St. Louis, MO, USA), NaOH (≥98%, Sigma-Aldrich, USA), silver nitrate (≥99%, Sigma-Aldrich, MO, USA), Palladium (II) nitrate solution (99.9%, Sigma-Aldrich, USA), Ni(NO_3_)_2_ 6H_2_O (99.99%, Sigma-Aldrich, USA), Sodium Borohydride (Fisher scientific, Waltham, MA, USA), Poly (vinyl alcohol) (PVA, 10,000, 80%, Sigma-Aldrich, MO, USA), and Nitrobenzene (≥99%, Sigma-Aldrich, USA). Deionized water was employed in all synthesis and this was supported by the Milli-Q direct 8 treatment system (Millipore, Molsheim, France). All chemicals were utilized as they were obtained, unless noted otherwise.

### 2.2. Synthesis

#### 2.2.1. LDH

A simple co-precipitation recipe was utilized to create an NiCr LDH with a 3:1 molar ratio, at a constant pH. Nickel nitrate and chromium nitrate (molar ratio 3:1) were dissolved in 100 mL H_2_O under vigorous agitating at ambient temperature. The solution of 2 M NaOH was then introduced dropwise into the Ni/Cr ions solution under a vigorous agitation until the pH attained was 9. The obtained precipitate was then transferred to a hydrothermal reactor vessel and heated at 80 °C for about 24 h. After cooling, via centrifugation, the obtained precipitates were separated and rinsed many times using DI water. Finally, the produced solid was dried at 80 °C for 12 h [29,30].

#### 2.2.2. Synthesis of Ag-Based and Pd-Based NiCr LDH

Typically, Pd(NO_3_)_2_ (6 mg/mL) or AgNO_3_ (6 mg/mL) aqueous solution are utilized to synthesize 1 wt.% of Ag NPs or 1% Pd NPs catalyst. After agitating the Pd or Ag solutions in deionized water for 30 min, polyvinyl alcohol (1 wt.%) was poured into the solution. A further addition of NaBH_4_ (0.2 M) was then introduced to the blend, resulting in a dark brown solution that was kept agitated for 40 min. Finally, the pre-prepared NiCr LDH solid was added to the mixture and agitated for an extra 60 min. The obtained nanocomposite was separated through filtration, rinsed with deionized water, and dried at 110 °C for 12 h. The fabricated nanocomposites were labeled as Ag@NiCr LDH and Pd@NiCr LDH [31].

### 2.3. Characterizations

X-ray diffraction (XRD) was exploited to explore the crystalline phases and the average crystalline sizes of the as-synthesized materials. An X-ray diffractometer (Shimadzu-D/Max2500VB2+/Pc) fitted with Cu Kα radiation (λ = 1.54056 Å, 40 kV, 35 mA) was utilized. The mean crystallite size of the synthesized nanocomposites was measured using the Debye–Scherrer equation as described in Equation (1):(1)$$D_{\left(\mathrm{particle}\;\mathrm{size}\right)}=\frac{\lambda \alpha }{\beta \cos\theta }$$
where θ is the diffraction’s angle, β is the full-width of 1/2 maximum (FWHM), D is the NP’s diameter, λ is the y wavelength (1.54056 Å), and α corresponds to the Scherrer constant (0.9). According to Equation (2), the basal spacing is derived from Bragg’s equation utilizing the (003) basal plane reflection [32]:(2)$$d(\mathrm{hkl})=\frac{n\lambda }{2\sin\theta }$$
where λ is the wavelength, n is a positive integer, and θ is the angle of scattering. Because of the hexagonal structure of NiCr LDH [33], the lattice parameters ‘a’ and ‘c’ were disclosed in α = β = 90°; γ = 120°; and a = b ≠ c, using Equation (3) [34].
(3)$$\frac{1}{d_{\left(\mathrm{hkl}\right)}^{2}}=\frac{4}{3}\;\left(\frac{h^{2}+\mathrm{hk}+k^{2}}{a^{2}}\right)+\frac{l^{2}}{c^{2}}$$

The “a” and “c” lattice parameters were determined using (110) and (003) facets, respectively. Fourier-transform infrared (ATR-FTIR) spectroscopy was explored utilizing the Shimadzu IR (Tracer-100 Fourier Transform) instrument, spanning the range from 400 to 4000 cm^−1^. The X-ray photoelectron spectroscopy (XPS) was disclosed utilizing Thermo-Fisher Scientific (K-ALPHA, USA) with Al Kα (200 eV and 50 eV). The surface features of the as-prepared nanocomposites were measured utilizing N_2_ sorption isotherms on a NOVA 4200e (Quantachrome Instruments, Florida, USA) at 77 K using the Brunauer–Emmett–Teller (BET) and the Barrett–Joyner–Halenda (BJH) approaches. The surface of the materials was prepared for measurement by subjecting them to activation through degassing at 120 °C for a duration of 12 h. The nanocomposite’s morphology was disclosed using field emission scanning electron microscopy (Zeiss FESEM Ultra 60) and high-resolution transmission electron microscopy (HRTEM, JEOL-2011, 200 kV).

### 2.4. Catalytic Performance Assessment

#### 2.4.1. Reduction of Nitrobenzene

To investigate the reduction capabilities of the as-synthesized catalysts, the nitrobenzene (NB) reduction employing NaBH_4_ was chosen as the model reaction. Typically, 5 mL of distilled water, 0.8 mL of 1.8 × 10^−4^ M NB, and NaBH_4_ (5 mg) dissolved in 1 mL of ice water are introduced into the glass tube. After adding 1 mg of Pd@NiCr LDH catalyst to this solution, a vortex was used to agitate it. A UV-Vis spectrometer (Agilent Cary 60) was used to monitor the reduction process and measure the reaction’s progress at various time intervals (1–8 min) [35]. The catalysts’ abilities to reduce nitrobenzene catalytically were assessed using the following Equations:(4)$$\mathrm{Conversion}\;(\%)=\frac{C_{o}-C_{t}}{C_{o}}\;\times \;100$$
(5)$$\ln[C_{t}]/[C_{o}]=-\mathrm{kt}$$
where C_o_ and C_t_ are the initial and the concentration of NB at a certain time (t) and the beginning, respectively. The rate constant (k) was determined from the slope of the linear graph of ln (C_t_/C_o_) against time (t).

#### 2.4.2. Carbon Monoxide Oxidation

The catalytic oxidation of carbon monoxide was examined using 20 mg catalyst that was placed in a quartz tube reactor. It was decided to use a programmable tube furnace to house the glass tube. The catalyst’s temperature was monitored utilizing a thermocouple that was positioned nearby [36]. The percentages of O_2_ and CO in helium gas are 20% and 4%, respectively. An MKS flowmeter was used to control the gas flow (50 cm^3^ min^−1^) and space velocity of 15,000 mL h^−1^ g^−1^. An infrared gas analyzer was utilized to reveal the rate of CO oxidation. To activate the materials before the measurements, they were first heated to 120 °C in He gas.

## 3. Results and Discussion

### 3.1. Spectroscopic Feature of Nanocomposites

#### 3.1.1. Crystal Structure

The FT-IR spectra of fabricated catalysts (NiCr LDH, Ag@NiCr LDH, and Pd@NiCr LDH) are depicted in Figure 1a. The wide band noted at 3280–3678 cm^−1^ is attributed to the stretching mode of the OH moieties and the intercalated molecules of H_2_O. In all samples, the bending mode of intercalated H_2_O molecules (δ H_2_O) is attributed to the feeble peak at 1636 cm^−1^. The NO_3_^−^ moieties in the LDH interlayer exhibit a stretching mode, which is connected to the band at 1339 cm^−1^ and was switched to 1343 cm^−1^ after decoration with Ag or Pd NPs due to changes in the electronic environment. Chemical bond modes of Cr-O, Ni-O, O-Cr-O, and O-Ni-O are ascribed to the characteristic stretching and bending vibrations at 511–700 cm^−1^ [37]. These results supported the fabrication of LDH. Furthermore, the NO_3_^−^ stretching band’s intensity decreases with a switch to a greater wavenumber when Ag or Pd NPs are immobilized on the NiCr LDH surface. This proved that the anion exchange is most efficient when NO_3_^-^ anions are involved, due to their ability to rotate within the LDH during the anion exchange more quickly than other kinds of anions [38].

To further explore the structure of the as-synthesized LDHs’ nanocomposites, XRD patterns were examined. Figure 1b depicts the XRD patterns of fabricated catalysts Ni-Cr LDH, Ag@NiCr LDH, and Pd@NiCr LDH. The synthesized NiCr LDH matched well to the LDH featured by the existence of base diffraction bands at 12.3°, 22.4°, 35.2°, 39.6°, 49.9°, 60.8°, and 61.4°, which can be ascribed to the features (003), (006), (012), (015), (018), (110), and (0015), which are facets of the layered double hydroxide. Furthermore, the relative feeble intensity and broader diffraction peaks disclose that LDHs have a low degree of crystallinity and thin two-dimensional structures. These findings unequivocally demonstrate that the NiCr LDH was successfully synthesized (JCPDS PDF-96-210-2794). After the decoration with Ag and Pd metallic NPs, the NiCr LDH’s XRD pattern maintained its characteristic structure, while the peaks widened and became a little weaker. Ag and Pd metallic NPs are widely dispersed across the LDH surface, as shown by the absence of any extra diffraction peaks associated with their crystalline phases. Additionally, the interlayer distance of 0.7169 nm could be estimated from the (003) diffraction peak at 12.33°, showing that H_2_O and NO_3_^−^ molecules could be intercalated between the layers rather than being on the surface (Table 1). Interestingly, upon loading Ag or Pd metallic NPs, the interlayer distance expanded to 0.7632 nm, which is explained by the incorporation of the metallic NPs between the LDH layers. The estimation of the lattice parameters (a) and (c) of the prepared NiCr LDH have been evaluated from the planes (003) and (110) by the following equations (a = 2d_110_ and c = 3d_003_) (Table 1). The average crystallite size for NiCr LDH, Ag@NiCr LDH, and Pd@NiCr LDH was estimated to be 10.1 nm, 11.2 nm, and 10.7 nm, respectively (Table 1).

#### 3.1.2. Surface Chemical State

To explore the surface chemical oxidation states of the obtained samples, XPS was additionally conducted. Ni^2+^ ions may be present in NiCr LDH, as seen in Figure 2A, where two distinct bands of Ni 2p at 856.3 and 873.8 eV are connected to the binding energies of Ni 2p_3/2_ and Ni 2p_1/2_, respectively [39]. In the spectrum of Cr 2p (Figure 2B), two wide peaks present at 577.6 and 586.7 eV were related to the Cr 2p_3/2_ and Cr 2p_1/2_, respectively. The band at 577.6 eV is associated with Cr^3+^ (OH), that also raises the possibility of trivalent Cr ions being present. The binding energy of Cr 2p at 576.3 and 577.6 eV indicates the development of the Cr–O and Cr–OH bond, respectively [40]. The estimates of the binding energy of the O 1s (Figure 2C) are 531.1 eV, 532 eV, and 532.5 eV, ascribed to the feature’s oxygen bands in, respectively, O_2_^-^, M-OH, and oxygen from bounded H_2_O. Therefore, the XPS findings verify the as-synthesized NiCr LDH.

The deconvoluted XPS spectrum of Ag 3d is illustrated in Figure 2D. Using the peak-differentiating approach, four bands at 367.4, 374.6, 366.5, and 373.4 eV are disclosed. In particular, two peaks at 367.4 and 374.6 eV are assigned to metallic Ag(0), and the bands at 366.5 and 373.4 eV are associated with Ag(I), implying that both Ag(0) and Ag(I) exist in the LDH simultaneously. In the Pd 3d shown in Figure 2E, the synthesized Pd@NiCr LDH exhibits two deconvoluted bands, Pd 3d_5/2_ and Pd 3d_3/2_. The low energy Pd 3d_5/2_ is resolved into Pd(0) and Pd(II) designated doublets at 336.1 and 338.4 eV. The Pd 3d_3/2_ does, however, produce two bands at 341.3 and 343.5 eV that are connected to the existence of Pd^2+^ and Pd° ion pairs. It is determined that Pd (0) has an atomic surface concentration of 86.1%, whereas Pd has a concentration of 13.9% (II).

### 3.2. Surface Characteristics and Porosity

The surface characteristics and porosity of the LDH samples were disclosed by nitrogen sorption isotherms at 77 K as depicted in Figure 3 and Table 2. It is obvious from the isotherm curves (Figure 3) that all of the samples display isotherm type IV and H3 hysteresis loops according to IUPAC classification, which are features of substances with plate-like particles and mesoporous structure (Figure 3). The desorption branch is probably dependent on connectivity, which is thought to be caused by capillary condensation and mesopores. Furthermore, it was discovered that there is a small adsorption–desorption hysteresis loop with the same mesoporous structure after decorating with Ag or Pd NPs. Table 2 presents the findings of surface characteristics. The BET surface area of the fabricated samples declined in the obtained rank: Pd@NiCr LDH (110.7 m^2^ g^−1^) > Ag@NiCr LDH (92.3 m^2^ g^−1^) > NiCr LDH (72.4 m^2^ g^−1^). The average diameters and volumes of the pores for all of the aforementioned samples were concurrently 12.3 nm and 0.3389 cm^3^, 7.4 nm and 0.2542 cm^3^, and 6.2 nm and 0.1562 cm^3^, respectively. The samples’ BJH average pore diameters ranged from 2 to 50 nm (inset in Figure 3), further demonstrating the presence of mesoporous structures. In contrast to the pure NiCr LDH, the Pd@NiCr LDH had a greater surface area and pore volume. This could be attributed to these nanoparticles providing additional active centers for catalytic processes and increasing the porosity of the material, leading to a higher surface area. Furthermore, the expansion that occurred when the metallic and bimetallic nanoparticles intercalated within the LDH interlayer as well as the intercalation of NPs hindered the agglomeration of NiCr LDH sheets, resulting in larger surface areas. The increased area of Pd@NiCr LDH and Ag@NiCr LDH simultaneously provided more active sites and improved catalytic efficacy.

### 3.3. Morphology Assessment

The LDHs’ structures of the prepared solids were additionally verified by the aid of FESEM-EDX and elemental mapping as depicted in Figure 4. It is obviously noted in all of the samples (Figure 4) that only one type of morphology, plate-like architecture, can be developed. The plate-like morphology with a smooth surface was noted due to the intercalation of the metallic nanoparticles which hindered the agglomeration of the LDH layers with the formation layer–layer architecture. Additionally, the elemental mapping and the associated EDX assessment (Figure 4) reveal the chemical compositions of the fabricated samples and unequivocally demonstrate that the Ni, Cr, O, Ag, and Pd elements were uniformly immobilized on NiCr LDH.

Figure 5A,B depicts the TEM, the HRTEM images, and the particle size distribution of Ag@NiCr LDH and Pd@NiCr LDH. The plates of all of the samples are stacked together with each other as illustrated in Figure 5. On the contrary, the TEM images of the NiCr LDH loaded with Ag or Pd NPs demonstrate that the metallic NPs have been immobilized, as is evident from the gradual change in color from faint gray to black. The lattice spacing that is attributed to the LDH reflection planes is shown on the HRTEM images. The histogram discloses that the distribution histogram reveals a mean size of 6.3 nm for Ag@NiCr LDH and 2.65 nm for Pd@NiCr LDH. In conclusion, it can be said that highly dispersed Ag or Pd NP-based catalysts were successfully developed utilizing NiCr LDH.

### 3.4. Catalytic Performance

#### CO Oxidation

Carbon monoxide, a toxic and potentially deadly gas, is created when fuels such as natural gas, coal, and gasoline are not fully burned. It has no odor or color and can bind to hemoglobin in blood cells. Large amounts of CO are regularly eliminated from sources such as power plants, transportation, and industrial and domestic activities. One solution to reduce carbon monoxide in the air is to convert it into CO_2_ through an oxidation process. The design of catalysts that facilitate this process at a sufficient rate is crucial for effectively removing carbon monoxide. Recently, this reaction has gained significant interest owing to its importance in areas such as automotive exhaust treatment, gas sensors, air purification, and fuel cells [41]. Figure 6 discloses the results of the catalytic efficacy of LDH samples toward CO oxidation. The temperatures at which 50% and 100% conversions occur were also determined. It is clear that the pure NiCr LDH does not have significant catalytic activity until 350 °C, suggesting that the observed conversion temperature is likely due to the synergistic effect between chromium and nickel oxides, rather than the original LDH [42]. Therefore, utilizing the synergistic effects between different ionic elements, as determined by XPS findings (Ni^2+^ and Cr^3+^/Cr^6+^), it is possible to enhance the catalytic activities [43]. It is noteworthy that the addition of 1 wt.% of Ag or Pd to the LDH greatly improves CO oxidation, as depicted in Figure 6. Specifically, the light-off temperature T_50%_ and T_100%_ of 1 wt.% Ag@NiCr LDH is reduced to 257.2 and 370.3 °C, respectively. Similarly, the light-off temperature T_50%_ and T_100%_ of 1 wt.% Pd@NiCr LDH decreases to 136.1 and 200.2 °C. The exceptional efficacy of the Pd@NiCr LDH catalyst can be ascribed to the robustly distributed Pd NPs on the LDH and their ability to supply active oxygen [44,45,46].

### 3.5. Nitrobenzene (NB) Reduction

The effectiveness of LDHs’ nanocomposites in catalyzing the reduction of nitrobenzene (NB) to aniline (AN) was evaluated using NaBH_4_ as the reducing agent. UV-vis absorption spectra that change over time can be used to easily track the progress of the reaction, as disclosed in Figure 7. The combination of NB with NaBH_4_ has a noticeable band at 270 nm when there are no catalysts present. The absorption peak at 270 nm remained constant for a prolonged period in the absence of a catalyst, indicating that NaBH_4_ is unable to reduce the 4-nitrophenolate ion without the presence of a catalyst. However, when catalysts are added, the peak at 270 nm caused by NB rapidly decreases, while the peak at 230 nm caused by the reduced product of AN increases. Since the amount of NaBH_4_ is much greater than that of NB, the pseudo-first-order model was applied to measure the catalytic rate constant and t_1/2_ using Equation (5). The reduction of NB to AN was started by adding catalysts to the mixture, as depicted in Figure 7a–c. As the catalytic reaction occurs, the peak of NB absorption at 270 nm decreases. It was observed that the Pd@NiCr LDH catalysts fully convert NB to AN in just 8 min, as shown in Figure 7c. This efficient performance is likely due to the well-dispersed Pd NPs over the LDH.

To grasp a deeper insight of the performance of the synthesized catalysts, kinetic assessments were carried out using the Langmuir–Hinshelwood model (L-H) [47]. Because the concentration of NaBH_4_ solution is high enough relative to the concentration of NB in the reaction, it could be assumed that the content remains constant throughout the catalytic reduction process. As a result, the reduction kinetics could be disclosed utilizing a pseudo-first-order equation to determine the reduction rate constant (k_app_). The graph in Figure 8a illustrates the correlation between ln(C_t_/C_o_) of NB and time, when using Ni-Cr LDH, Ag@NiCr LDH, and Pd@NiCr LDH catalysts. Figure 8b illustrates the percentage of NB that is converted to AN at different reaction times.

From Figure 8a, it could be disclosed that the ln C_t_/C_o_ vs. time graph depicts a strong linear relation (R^2^~0.9134–0.9815). The k_app_ at ambient temperature for various catalysts were estimated. The heterogenous rate constants of NiCr LDH, Ag@NiCr LDH, and Pd@NiCr LDH nanocomposites were measured to be 0.0339 min^−1^, 0.0528 min^−1^, and 0.3648 min^−1^, respectively, which discloses the best performance for converting NB to AN for the Pd@NiCr LDH nanocomposite compared with other nanocomposites (Table 3). It is worth mentioning that the Pd@NiCr LDH catalyst has the best performance, taking only 8 min to achieve 98.7% conversion, as depicted in Figure 8b. On the other hand, the pure NiCr LDH has the lowest performance (46.2%) and is unable to fully convert NB even within 30 min. The increased catalytic performance is due to the close proximity and interaction between the prepared NPs’ catalysts and nitrobenzene.

The higher activity of silver (Ag) and palladium (Pd) samples can be held accountable for their distinctive electronic and catalytic properties. Ag has a relatively low electron affinity, which makes it highly reactive with other elements and compounds, contributing to its high activity. Additionally, Ag has a robust surface area and a high degree of surface reactivity, which further contributes to its activity. Pd, on the other hand, has exceptional catalytic properties. It has a unique electronic structure that allows it to easily form bonds with other elements, making it an efficient catalyst for a wide variety of chemical processes. This is due to the presence of d-electrons in its electron configuration, which makes it highly reactive. Furthermore, Pd has a high surface-area-to-volume ratio, which increases its effectiveness as a catalyst. Additionally, Pd is an effective catalyst in both acidic and basic conditions, making it versatile in a variety of chemical reactions. Its ability to catalyze oxidation and hydrogenation reactions, as well as its resistance to poisoning by impurities, further contribute to its high activity.

## 4. Conclusions

In summary, a facile hydrothermal coprecipitation approach followed by a sol immobilization approach was used to synthesize various bifunctional catalysts (NiCr LDH, Ag@NiCr LDH, and Pd@NiCr LDH) for the NaBH_4_-mediated nitrobenzene reduction to aniline and CO oxidation, which disclosed that the Pd@NiCr LDH catalyst not only displayed robust activity towards nitro compounds’ reduction but also improved the performance of the oxidation of carbon monoxide for air purification.

## Figures and Tables

**Figure 1 nanomaterials-13-00987-f001:**
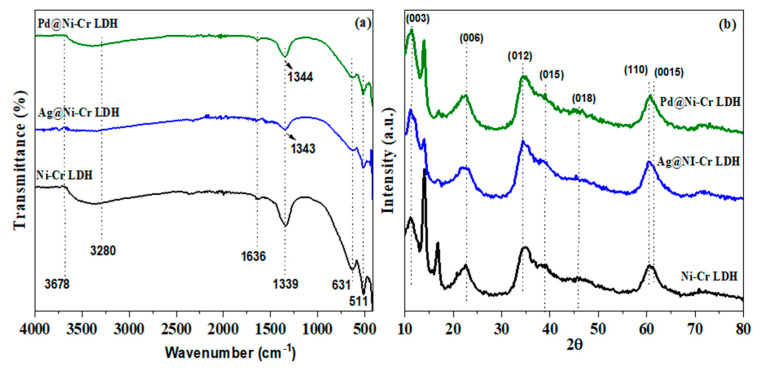
(**a**) FTIR; (**b**) XRD diffraction patterns of NiCr LDH, Ag@NiCr LDH, and Pd@NiCr LDH nanocatalysts.

**Figure 2 nanomaterials-13-00987-f002:**
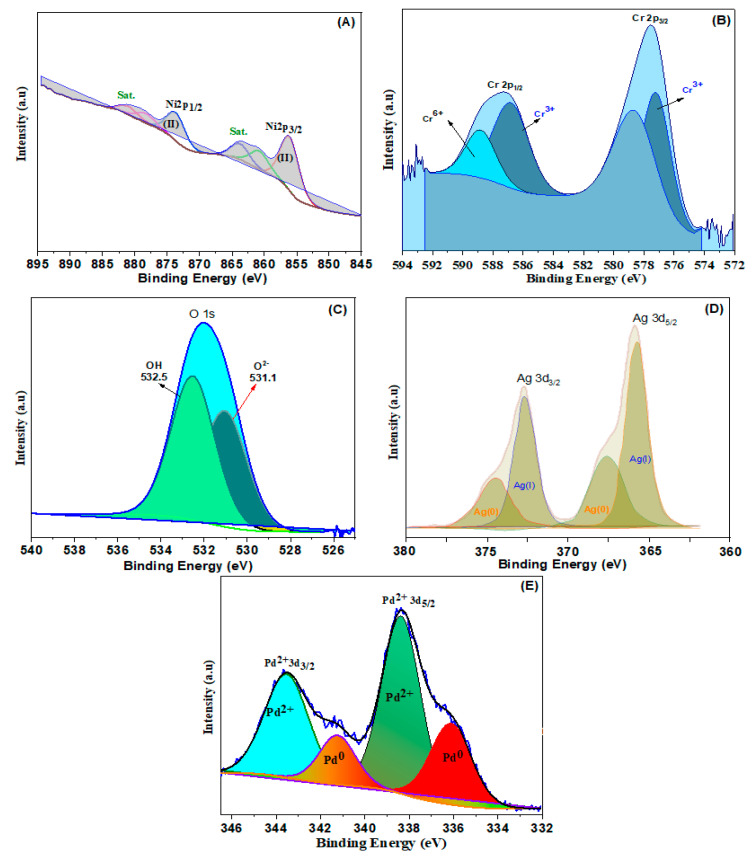
High resolution of (**A**) Ni 2p, (**B**) Cr 2p, (**C**) O 1s, (**D**) Ag 3d, and (**E**) Pd 3d spectra of LDHs.

**Figure 3 nanomaterials-13-00987-f003:**
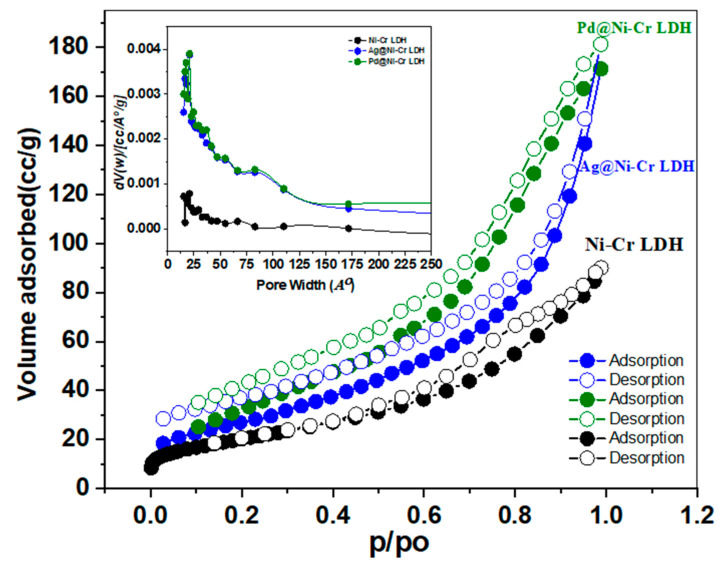
Nitrogen sorption isotherms of NiCr LDH, Ag@NiCr LDH, and Pd@NiCr LDH nanocomposites (inset illustrates the pore size distribution of the same samples).

**Figure 4 nanomaterials-13-00987-f004:**
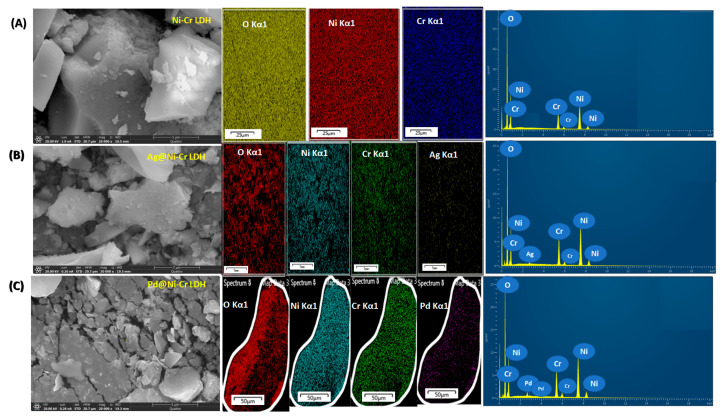
FESEM images, mapping images of the elemental distribution, and EDX analysis of (**A**) NiCr LDH, (**B**) Ag@NiCr LDH, and (**C**) Pd@NiCr LDH nanocatalysts.

**Figure 5 nanomaterials-13-00987-f005:**
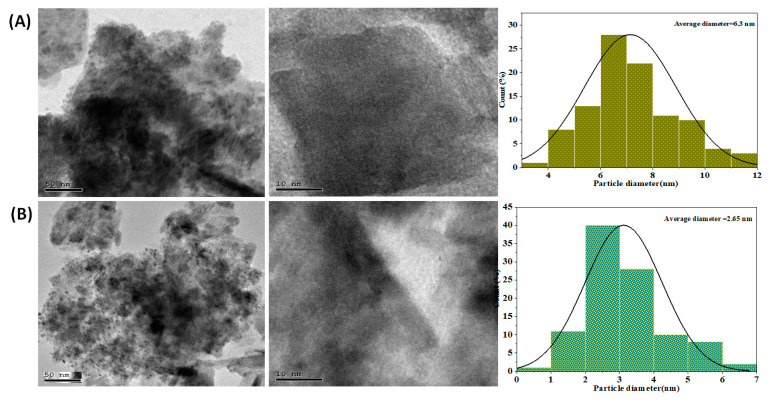
TEM, HRTEM images, and particle size distribution of (**A**) Ag@NiCr LDH, (**B**) TEM, and HRTEM images of Pd@NiCr LDH.

**Figure 6 nanomaterials-13-00987-f006:**
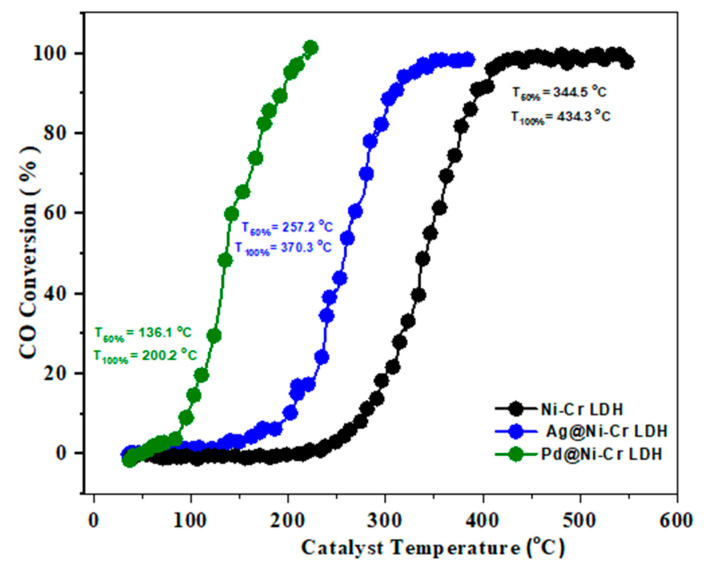
Light-off curves for CO oxidation over NiCr LDH, Ag@NiCr LDH, and Pd@NiCr LDH nanocatalysts.

**Figure 7 nanomaterials-13-00987-f007:**
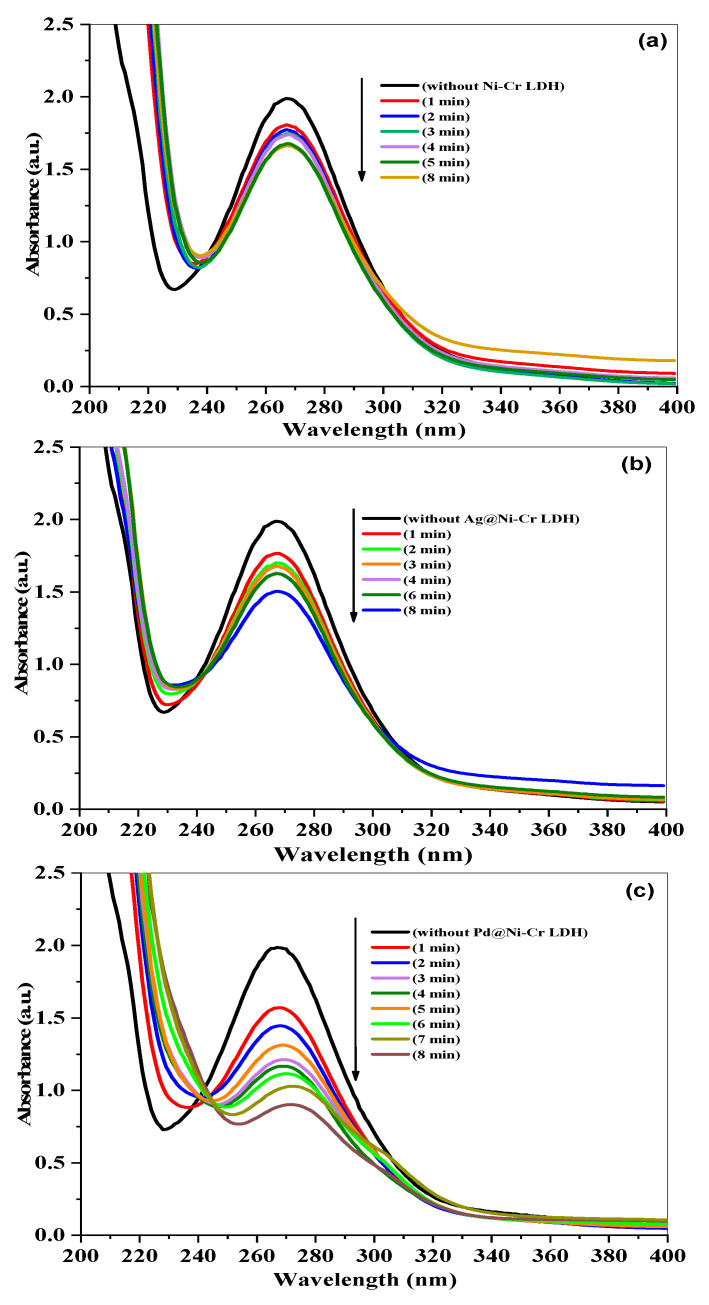
Time-dependent variations in absorption spectrum of NB reduction by NaBH_4_ in the presence of (**a**) NiCr LDH, (**b**) Ag@NiCr LDH, and (**c**) Pd@NiCr LDH nanocatalysts.

**Figure 8 nanomaterials-13-00987-f008:**
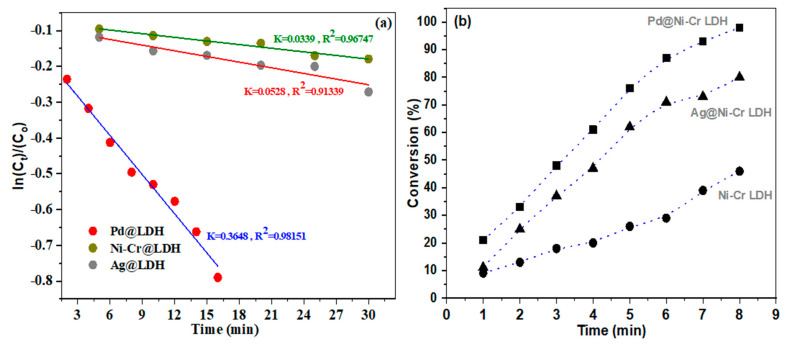
(**a**) Plots of ln(C_t_/C_o_) of NB versus time and (**b**) conversion (%) of NB as a function of working time reduction by NaBH_4_ in presence of NiCr LDH, Ag@NiCr LDH, and Pd@NiCr LDH.

**Table 1 nanomaterials-13-00987-t001:** Structure variables of Ni-Cr LDHs evaluated from XRD diffraction patterns.

Catalysts	d_(003)_/ (nm)	d_(006)_/ (nm)	d_(110)_/ (nm)	Lattice Parameter (nm)		Particle Size (nm)
				a = 2d_110_	c = 3d_003_	
NiCr LDH	0.7169	0.3964	0.1521	0.3042	2.1507	10.1
Ag@NiCr LDH	0.7202	0.3984	0.1521	0.3042	2.1606	11.2
Pd@NiCr LDH	0.7632	0.3921	0.1522	0.3044	2.2893	10.7

**Table 2 nanomaterials-13-00987-t002:** Surface characteristics of various NiCr LDH samples.

Catalysts	S_BET_/m^2^ g^−1^	D_BJH_/nm	V_BJH_/cm^3^ g^−1^
Ni-Cr LDH	72.4	6.2	0.1562
Ag@Ni-Cr LDH	92.3	7.4	0.2542
Pd@Ni-Cr LDH	110.7	12.3	0.3389

**Table 3 nanomaterials-13-00987-t003:** The conversion (%) and kinetic parameters for the reduction of nitrobenzene over various Noria-GO nanocomposites.

Catalyst	Conversion (%) *	Kinetic Parameters			
		K_app_ (min^−1^)	R^2^	t_1/2_ (min)	k_AF_ ** (min^−1^ g^−1^)
NiCr LDH	46.2	0.034	0.9675	20.4	34
Ag@ NiCr LDH	78.3	0.053	0.9134	13.1	53
Pd@ NiCr LDH	98.7	0.365	0.9815	1.9	365

The reaction includes 1 mg catalyst and 5 mg of NaBH_4_. * The conversion provided was calculated at the reaction time (8 min). ** The activity factor (k_AF_) was calculated using k_AF_ = k_app_/catalyst mass (g).

## Data Availability

The corresponding author can provide access to the data presented in this study upon request.

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
