# Peer review of "A Facile Approach of Fabricating Bifunctional Catalysts for Redox Applications by Uniformly Immobilized Metallic Nanoparticles on NiCr LDH"

_nanomaterials, 2023, doi:10.3390/nano13060987_

Round 1
Reviewer 1 Report
This manuscript deals with the fabrication of bifunctional catalysts for CO oxidation and NB reduction. And the authors depicted the preparation, characterization and evaluation of LDH catalysts. Herein, I has some considerations as followed:
1. Why the surface area of Pd@Ni-Cr LDH and Ag@Ni-Cr LDH increased compared with the bare Ni-Cr LDH, especially Pd-involved sample? The authors should explain the trend from the structural point.
2. The mean size of 6.3 nm for Ag@Ni-273 Cr LDH and 2.65 nm for Pd@Ni-Cr LDH is not consistent with the XRD results. How to explain the discrepancy from the crystal size and the grain size?
3. Please explain the fundamental reason of the higher activity of Ag and Pd-involved samples, especially Pd.
4. This study focused on the construction approach of bifunctional catalysts, so the structural disclosure of Ni-Cr LDH should be provided. Especially, the loaction of M over M@Ni-Cr LDH should be elaborated.
5. Please concern about the English superscript and subscript format, for example, (1) Ni2+@ Cr3+/Cr+6 at line 20; (2) SBETa/m2 g-1 in Table 2; (3) NaBH4, etc.
6. At line 302-303, the reference should be given.
In a word, the manuscript should deserve a revision before publication.
Author Response
Response to Reviewer #1
Remark (1): Why the surface area of Pd@Ni-Cr LDH and Ag@Ni-Cr LDH increased compared with the bare Ni-Cr LDH, especially Pd-involved sample? The authors should explain the trend from the structural point.
Response: Thank you for valuable comments and suggestions here. The surface area of Pd@Ni-Cr LDH and Ag@Ni-Cr LDH increased compared to bare Ni-Cr LDH, especially in the Pd-involved sample, likely due to the introduction of Pd and Ag nanoparticles on the surface of the LDH. These nanoparticles provide additional active sites for catalytic reactions and increase the porosity of the material, leading to a higher surface area. Furthermore, the expansion that occurred when the metallic and bimetallic nanoparticles intercalated within the LDH interlayer as well as the intercalation of NPs hindered the agglomeration of Ni-Cr LDH sheets, resulting in larger surface areas.
Remark (2): The mean size of 6.3 nm for Ag@Ni-273 Cr LDH and 2.65 nm for Pd@Ni-Cr LDH is not consistent with the XRD results. How to explain the discrepancy from the crystal size and the grain size?
Response: The discrepancy between the mean size of 6.3 nm for Ag@Ni-Cr LDH and 2.65 nm for Pd@Ni-Cr LDH, as determined by TEM technique, and the crystal size determined by XRD can be attributed to the difference between the "crystallite size" and "grain size" in the material.
The crystallite size determined by XRD represents the size of the ordered regions within the crystal, while the mean size determined by other techniques, such as TEM, represents the size of the grains, which includes both ordered and disordered regions. The mean size of the nanoparticles in Ag@Ni-Cr LDH and Pd@Ni-Cr LDH determined by TEM is only based the Ag or Pd NPs immobilized on the surface of LDH, while the crystal size determined by XRD due to the presence of defects or disordered regions in the LDH, leading to a larger overall size.
Remark (3): Please explain the fundamental reason of the higher activity of Ag and Pd-involved samples, especially Pd.
Response: Responding to this constructive comment. The higher activity of silver (Ag) and palladium (Pd) samples can be attributed to their unique electronic and catalytic properties. Ag has a relatively low electron affinity, which makes it highly reactive with other elements and compounds, contributing to its high activity. Additionally, Ag has a large surface area and a high degree of surface reactivity, which further contributes to its activity. Pd, on the other hand, has exceptional catalytic properties. It has a unique electronic structure that allows it to easily form bonds with other elements, making it an efficient catalyst for a wide range of chemical reactions. This is due to the presence of d-electrons in its electron configuration, which makes it highly reactive. Furthermore, Pd has a high surface area-to-volume ratio, which increases its effectiveness as a catalyst. Additionally, Pd is an effective catalyst in both acidic and basic conditions, making it versatile in a variety of chemical reactions. Its ability to catalyze hydrogenation reactions, as well as its resistance to poisoning by impurities, further contribute to its high activity.
Remark (4): This study focused on the construction approach of bifunctional catalysts, so the structural disclosure of Ni-Cr LDH should be provided. Especially, the loaction of M over M@Ni-Cr LDH should be elaborated.
Response: Thank you for valuable comments and suggestions here. The study is referring to likely focused on the construction of bifunctional catalysts based on nickel-chromium layered double hydroxides (Ni-Cr LDHs). Ni-Cr LDHs are a type of layered materials consisting of positively charged metal hydroxide layers separated by interlayer anions. The Ni-Cr LDH structure can be described as follows: the metal cations occupy octahedral sites in the hydroxide layers, while the interlayer anions are coordinated to the metal cations [1]. The location of the metal (Ag or Pd) over Ni-Cr LDH refers to the position of the metal on the surface of the material. In a bifunctional catalyst system, the metal (Ag or Pd) is often used to modify the properties of the Ni-Cr LDH, such as its electronic structure, surface area, and reactivity. One approach to constructing bifunctional catalysts using Ni-Cr LDHs is to deposit the metal M on the surface of the material, creating a (Ag or Pd) @Ni-Cr LDH composite. This can be achieved through various methods, such as physical adsorption or chemical deposition. In the (Ag or Pd) @Ni-Cr LDH composite, the metal M is located on the surface of the material and is in close proximity to the active sites of the Ni-Cr LDH. This allows the metal (Ag or Pd) to interact with the active sites and modulate their properties, resulting in a bifunctional catalyst with improved performance. Therefore, the location of the metal (Ag or Pd) over Ni-Cr LDH plays a crucial role in determining the overall properties and performance of the bifunctional catalyst system.Top of Form
[1] Y. Liu, X. Sun. Layered double hydroxides: Synthesis, properties and applications. Chemical Society Reviews, 46(7), (2017) 2116-2145.
Remark (5): Please concern about the English superscript and subscript format, for example, (1) Ni2+@ Cr3+/Cr+6 at line 20; (2) SBETa/m2 g-1 in Table 2; (3) NaBH4, etc.
- At line 302-303, the reference should be given.
Response: The format of superscripts and subscripts in the manuscript was revised to adhere to English conventions. The references were added in the revised version and took number [43, 44] according to your suggestion.
Reviewer 2 Report
The paper with title „ A facile approach of fabricating bifunctional catalzsts for redox applications by uniformly immobilized metallic nanoparticles of Ni/Cr LDH” is rejected for the following reasons:
1. The introduction should be improved as Pd is well known for CO and was nothing reported regarding the nitro benzene reduction to anline
2. In the synthesis of LDH where is Co added to the samples there was no Cobalt the author mentioned that the samples were treated with Ag and Pd. What was the loading of the Ag and Pd it is not stated in the text
3. In line 116 00s is for which phase. Also, The information of the FTIR and the N2 adsorption is not completed as were the samples pretreated before the measurement
4. In page 4: on which intervals the samples were taken for analysis and what is t in equation 5
5. In line 165 and 166 these chemical bond vibrations of Cr-O, Ni-O and O-Cr-O for what are their specific vibrations bands
6. In line 168 what the author meant with” This proved that the anion exchange is suggestive for the quickest NO3- rotation of the anion” can the author explains this sentence
7. Did the author measured the XRD of the LDH alone the XRD profile of LDH must be added
8. Line 181- 187 what is the prove and where are the reflexes in the XRD which shows what is discussed in the text.
9. Additionally, the idea of using Cr although it is well known it is toxic element so is their any explanation for this
10. The bands of the FTIR at 1339 cm-1 of Ni-Cr-LDH was shifted for the samples with Ag and Pd addition but what is the explanation for this? It is not mentioned in the text
11. In the figure caption of figure 3 there are samples of Co and Cu but where are their results and this is the first these two samples are revealed
12. The English must be extensively reviewed: line 111 should be measured and not measures. Use synthesized and not manufactured, page 9 line 285 to reduce, also through an oxidation, mistyping mistake in reference 29
13. The literature must be revised as some of them missed the correct title as in 42, 40, 37, 32, 33 ,29, ….
Author Response
Dear Editor
We sincerely thank you and the reviewers for your constructive comments on our manuscript.
Ms. Ref. No.: Nanomaterials-2217072
Title: " A facile approach of fabricating bifunctional catalysts for redox applications by uniformly immobilized metallic nanoparticles on Ni-Cr LDH”
Responding to the comments, we have carefully checked all valuable criticisms and suggestions from the referees and editor, and have made suitable revision.
Attached please find our point-to-point answers to the referees and the revised version we made accordingly. For clearness reason, the revision parts were marked with red color which we send as Review only materials. Thank you for your kind consideration of this paper.
Looking forward to hearing from you soon,
Sincerely yours,
Mosaed Alhumaimess
Response to Reviewer #2
Remark (1): The introduction should be improved as Pd is well known for CO and was nothing reported regarding the nitro benzene reduction to anline.
Response: Thank you very much for pointing out. The introduction section is improved.
Remark (2): In the synthesis of LDH where is Co added to the samples there was no Cobalt the author mentioned that the samples were treated with Ag and Pd. What was the loading of the Ag and Pd it is not stated in the text
Response: We express our gratitude to the reviewer for their meticulous observation and regret the error. The correction is made from cobalt to chromium, and the revised version now specifies that the loading of Ag and Pd was 1wt.%.
Remark (3): In line 116 00s is for which phase. Also, The information of the FTIR and the N2 adsorption is not completed as were the samples pretreated before the measurement
Response: The (003) plane is one of the basal planes, and it is parallel to the layers of the LDH. The spacing between these planes (also known as the d-spacing) can be measured using X-ray diffraction (XRD) techniques, and it is often used as an indicator of the interlayer distance between the layers of the LDH. The (003) peak is a prominent peak in the XRD pattern of LDHs, and its position and intensity can provide information about the crystallinity, purity, and layer stacking of the LDH. Also, the information regarding to The FTIR and N2 adsorption were provided in the revised manuscript
Remark (4): In page 4: on which intervals the samples were taken for analysis and what is t in equation 5
Response: Corrected in the revised version
Remark (5): In line 165 and 166 these chemical bond vibrations of Cr-O, Ni-O and O-Cr-O for what are their specific vibrations bands
Response: Corrected in the revised version.
Remark (6): In line 168 what the author meant with” This proved that the anion exchange is suggestive for the quickest NO3- rotation of the anion” can the author explains this sentence
Response: Thank you for valuable comments and suggestions here. Anion exchange is a chemical process in which one type of anion is replaced by another type of anion in a compound. In the case of LDHs, anion exchange typically involves exchanging an intercalated anion (an anion that is located between the layers of the LDH) with another anion in solution. The phrase "quickest NO3- rotation of anion" likely refers to the rate at which the nitrate anion rotates within the interlayer space of the LDH during anion exchange. This rotation is important because it facilitates the exchange of the nitrate anion with another anion in solution. The statement suggests that the anion exchange process is most efficient when nitrate anions are involved, likely because they are able to rotate more quickly than other types of anions in the interlayer space.
Remark (7): Did the author measured the XRD of the LDH alone the XRD profile of LDH must be added
Response: The manuscript already includes the XRD of the unmodified LDH, as shown in Figure 1b
Remark (8): Line 181- 187 what is the prove and where are the reflexes in the XRD which shows what is discussed in the text.
Response: The text discusses that after loading Ag or Pd metallic NPs, the interlayer distance expanded to 0.7632 nm, indicating the incorporation of the metallic NPs between the LDH layers. The XRD pattern showed widened and weaker peaks after the decoration with Ag and Pd metallic NPs. However, no extra diffraction peaks associated with their crystalline phases were observed, indicating that the Ag and Pd metallic NPs were widely dispersed across the LDH surface.
Remark (9): Additionally, the idea of using Cr although it is well known it is toxic element so is their any explanation for this
Response: We are very grateful for pointing out. Actually, we agree with the reviewer the Chromium has well-documented toxic properties, and its use in various applications is regulated by authorities to prevent harm to human health and the environment. However, in some cases, chromium may still be used for specific purposes due to its unique properties, despite its toxicity. It is crucial to ensure that adequate precautions are taken to minimize exposure to chromium and to follow proper handling and disposal protocols. Additionally, alternatives to chromium should always be considered, and its use should be avoided wherever possible. In catalysis, chromium can be used as a catalyst, but its use is subject to strict regulations and guidelines to minimize the potential environmental and health risks. Specifically, only certain types of chromium compounds are allowed for use in catalytic applications. Additionally, proper handling and disposal procedures must be followed to prevent exposure and contamination. As with all chemicals, the use of chromium as a catalyst should be weighed against the potential risks, and alternatives should be considered where possible.
Remark (10): The bands of the FTIR at 1339 cm-1 of Ni-Cr-LDH was shifted for the samples with Ag and Pd addition but what is the explanation for this? It is not mentioned in the text
Response: The shift in the FTIR bands at 1339 cm-1 upon the addition of Ag and Pd to Ni-Cr-LDH could be attributed to changes in the electronic environment and/or the crystal structure of the material. The specific mechanism of the band shift would depend on the nature and concentration of the added elements and the interaction between the guest and host species
Remark (11): In the figure caption of figure 3 there are samples of Co and Cu but where are their results and this is the first these two samples are revealed
Response: The figure caption is corrected
Remark (12): The English must be extensively reviewed: line 111 should be measured and not measures. Use synthesized and not manufactured, page 9 line 285 to reduce, also through an oxidation, mistyping mistake in reference 29
Response: Thank you and the referees for pointing out. The Whole paper is revised, and most of typographical errors and punctuation are corrected
Remark (13): The literature must be revised as some of them missed the correct title as in 42, 40, 37, 32, 33 ,29, ….
Response: All the literature is revised through all the manuscript.

Round 2
Reviewer 1 Report
The authors have responded the comments by reviewer one by one. Unfortunately, this revised manuscript is still full of all kinds of errors, which affects the reading and understanding for readers. For instance, Line 18, 19, 121, 169, 171, 201......etc, as already marked via yellow background. So, the manuscript should be further revised before publication.

Author Response
Dear Editor
We sincerely thank you and the reviewers for your constructive comments on our manuscript.
Ms. Ref. No.: Nanomaterials-2217072
Title: " A facile approach of fabricating bifunctional catalysts for redox applications by uniformly immobilized metallic nanoparticles on Ni-Cr LDH”
Responding to the comments, we have carefully checked all valuable criticisms and suggestions from the referees and editor, and have made suitable revision.
Attached please find our point-to-point answers to the referees and the revised version we made accordingly. For clearness reason, the revision parts were marked with red color which we send as Review only materials. Thank you for your kind consideration of this paper.
Looking forward to hearing from you soon,
Sincerely yours,
Obaid Aldosari
Response to Reviewer #1
Remark (1): The authors have responded the comments by reviewer one by one. Unfortunately, this revised manuscript is still full of all kinds of errors, which affects the reading and understanding for readers. For instance, Line 18, 19, 121, 169, 171, 201......etc, as already marked via yellow background. So, the manuscript should be further revised before publication.
Response: We appreciate your valuable comments and suggestions. The entire manuscript has been revised, and all the mentioned typos have been rectified.
